# The Consequences of Delaying Telling Children with Perinatal HIV About Their Diagnosis as Perceived by Healthcare Workers in the Eastern Cape; A Qualitative Study

**DOI:** 10.3390/children7120289

**Published:** 2020-12-11

**Authors:** Sphiwe Madiba, Cynthia Diko

**Affiliations:** Department of Public Health, School of Health Care Sciences, Sefako Makgatho Health Sciences University, Pretoria 0001, South Africa; zithawellie51@gmail.com

**Keywords:** South Africa, rural, perinatal HIV, delayed disclosure, negative outcomes, children, caregivers

## Abstract

Although the benefits of disclosure are considerable, informing children with perinatal HIV of their own HIV status is often delayed to late adolescence. This study examined the social and contextual challenges that influence delaying disclosure to children and assessed the outcomes of delayed disclosure on the psychosocial health of children as perceived by the healthcare workers (HCWs) providing care to these children. Data were collected from HCWs via focus group discussions. Nurses, lay counsellors, social workers, and dieticians were selected from facilities in a rural South African health district. Thematic analysis was performed. The caregivers’ social context was the main barrier against informing children timely about their HIV diagnosis. The extent of the internalised HIV stigma influenced the delay in disclosing to the children. Delaying disclosure contributes to children’s refusing to take their medication, leads to the accidental disclosure of HIV, give rise to anger and resentment towards the caregiver, increase the risk of secondary transmitting of HIV, and poor health outcomes. It is essential to train HCWs to support caregivers and children through the disclosure process to ensure that caregivers realise the benefits of disclosure. Strategies to encourage caregivers to disclose early should be sensitive to their concerns about the negative impacts of disclosure.

## 1. Introduction

In many settings in sub-Saharan Africa (SSA), significant numbers of children and adolescents with perinatal human immunodeficiency virus (HIV) who regularly attend clinics and take antiretroviral treatment (ART) are not fully informed about their HIV diagnosis [1,2,3]. Disclosure of HIV status becomes crucial and beneficial as children approach puberty and need to transition to adult care. Timely and safe disclosure enhances their adherence to medication and their understanding of the disease and enables their active participation in self-care and treatment. HIV disclosure is also positively associated with safer sex behaviours in adolescents and increased access to social support [4,5,6,7,8,9,10]. Being aware of their own HIV status as they transition into adolescence is crucial for young people living with HIV in order that they may assume some responsibility for managing their own treatment [11].

Although the benefits of disclosure are considerable, informing a child of his or her own HIV status is often delayed. Consequently, in SSA, disclosure to children with perinatal HIV (PHIV) ranges from 13% to 60% [11,12,13]. While the World Health Organization’s (WHO) recommended age to complete the disclosure process to children with PHIV is before the age of 12 years [14], existing data indicate that the disclosure process often occurs late in the adolescent period [12,15]. For example, the median age at disclosure was 15 years in a study conducted in Kinshasa [16], 15 years in West Africa [15], 14 years in Nigeria [17], and in Kenya, disclosure occurred between the ages of 13 and 16 years [18]. As a result, the prevalence of disclosure to younger children is much lower than that of older children in SSA [13].

Although healthcare workers (HCWs) and caregivers believe disclosure to be important, disclosure is a complex process for caregivers and significant barriers prevent or delay disclosure from occurring. These factors influence how and when caregivers decide to disclose to a child [19,20,21,22]. Several studies have identified the various caregiver- and child-related factors that influence early and timely initiation of disclosure [8,11,13,23,24,25]. Caregivers delay the initiation of disclosure to late adolescence because they believe that when the child is too young he/she would not understand HIV-related information [2,12,13,26].

There are several caregiver-related factors that influence delaying timely disclosure. In many settings in SSA, disclosure is particularly difficult because of the HIV-related stigma, and caregivers delay disclosure for fear of subsequent stigma and discrimination. The fear of disclosure to other people because of stigma will continue to hinder the disclosure of HIV serostatus to children [26]. Feeling unprepared to answer the child’s questions and lacking disclosure skills or knowledge of how to disclose have been identified as major barriers to the early initiation of disclosing to children [11,22,23,24,27,28,29].

As such, many caregivers require assistance from HCWs to prepare them to initiate and facilitate disclosure to their children [27,28]. Beima-Sofie et al. [30] suggest that the lack of caregiver readiness has redefined the role of HCWs to that of educating, empowering, and persuading caregivers about the importance of HIV disclosure.

Whereas HCWs are central to the disclosure process, few studies have focussed on the perspectives and experiences of HCWs on the practice of disclosure [8,31] and even fewer studies have focussed on the psychosocial issues among adolescents as a result of delayed disclosure of HIV [3,32]. While disclosure has substantial benefits, delayed disclosure can accentuate emotional and behavioural disorders and might lead to unintended negative psychological outcomes in children and adolescents with PHIV [9,13,15,25,33,34]. In addition, the social and contextual challenges that influence disclosure to children with PHIV have not been extensively explored [7,12,28].

The purpose of this study was to examine the social and contextual challenges that influence delayed disclosure to children with perinatal HIV and to assess the outcomes of delayed disclosure on the psychosocial health of children and adolescents as perceived by HCWs providing HIV treatment and care to these children. It is important to encourage the culturally appropriate, safe, and timely disclosure of HIV status to children and adolescents as a central principle in the care and management of these children [35].

## 2. Materials and Methods

### 2.1. Design of the Study

The study was conducted in two sub-districts of Alfred Nzo District in the Eastern Cape Province of South Africa. Alfred Nzo is the smallest district in the Eastern Cape Province, predominantly rural with an estimated total catchment population of 866,646. About 10% of the adult population in this District is infected with HIV, representing 11% of the total population of the Eastern Cape living with HIV estimated at 25.2%. There are 72 eight-hour clinics, two community health centres (CHCs), six district hospitals, and one TB hospital providing services to the population. At the time of the fieldwork, the district had about 6354 children under 15 years enrolled in the HAART programme, which is 7.2% of the total population living with HIV in the district. The setting of the study was 23 health facilities. The database of children receiving ART was used to select the four hospitals, one CHC and eighteen clinics with the greatest number of children with PHIV.

The study utilised purposive sampling to recruit HCWs to participate in focus group discussions (FGDs) between April and September 2018. HCWs were recruited for FGDs by the lead researcher in collaboration with the facility managers of the selected facilities. Given the small number of nurses who provide HIV services at these facilities, all HCWs who were involved in offering HIV care services to children and adolescents with PHIV for more than six months in each of the selected facilities were invited to participate. Sampling was done to achieve diversity of the sample characteristics and the study sample consisted of nurses, social workers, dieticians, and lay counsellors.

### 2.2. Data Collection

Eight focus group discussions were held with a total of 51 HCWs. The discussions used a semi-structured guide with open-ended questions. The guide was developed after an extensive review of the existing studies on the disclosure of HIV status to children and adolescents, and was further adapted from a previous study conducted by the first author and a colleague [8]. The guide asked open-ended questions pertaining to the HCWs’ opinions regarding the contextual barriers to disclosing HIV diagnosis to children with PHIV, their beliefs about who should tell the child about his/her HIV diagnosis, their perceptions about the ideal age of disclosing to children, and the outcome of delaying disclosure to children. Follow-up questions and probes encouraged the participants to express in detail their experiences and perspectives of disclosing to children. The FGDs were moderated by the lead researcher (CD) and a research assistant who was recruited and trained in the conduct of focus group discussions. The first author (SM) supervised the fieldwork throughout the research project.

The FGDs were conducted at the selected health facilities in a private room allocated by the facilities to ensure privacy. The discussions were conducted in English and IsiXhosa, the local language, to allow the participants to respond in the language that they were comfortable with for maximum participation. Each focus group discussion lasted for about 60 min and was audio-recorded with the consent of the participants. Additional field notes were taken.

### 2.3. Data Analysis

Audio recordings of the FGDs were transcribed verbatim by CD and the research assistant. CD, who is fluent in the languages, verified the transcription by re-reading the transcribed data while listening to the recorded data. CD was involved in the transcription as a way of familiarising herself with the data. Both authors were involved in the data analysis and analysed the English-language transcripts using NVivo™ Version 10 software, following a thematic approach as described by Braun and Clarke [36] using both inductive and deductive approaches. The inductive approach was used to identify codes and meanings that emerged from within the data, and in the deductive approach, a priori codes from the focus group guide were used. Analysis began with the authors independently reading a few of the transcripts repeatedly to familiarise themselves with the data and identify initial emerging codes. Next, they searched for statements of meaning, and identified codes and emerging themes across the transcripts as part of the process that formed a framework for a codebook. After consensus about the definitions of themes had been reached by the authors, the codebook was finalised. Once the codebook had been completed, the authors applied coding to the remaining transcripts, and analysis continued until deep and rich themes and subthemes had been arrived at. The final themes and sub-themes were decided upon by agreement between the authors.

Rigour in qualitative research ensures that the findings of the study are credible and therefore transferable to other settings in similar contexts [37]. Credibility, dependability, transferability, and conformability are strategies used to attain rigour [38]. The triangulation of data sources, keeping an audit trail of the research processes and procedures from the inception of the study are strategies that were used to attain trustworthiness in the study. In addition, both authors conducted the analysis and were immersed in the data to reduce investigator bias, and maintained an audit trail of all the data analysis activities [39].

### 2.4. Ethical Considerations

The protocol was approved by the Research and Ethics Committee of Sefako Makgatho Health Sciences University (SMUREC/H/225/2017: PG). Permission was also obtained from the Eastern Cape Department of Health Research Ethics Committee to access the selected facilities. All the participants provided written consent before the GFDs took place.

## 3. Results

### 3.1. Description of the Study Participants

Demographic features of the HCW participants are summarised in Table 1. Data were collected from eight FGDs with a total of 51 HCWs of whom 46 were females whilst 5 were males. Their ages ranged between 22 and 68 years with a mean age of 43.4 years. The sample was comprised of 24 nurses, 16 lay counsellors, 5 social workers, 3 dieticians, and 3 enrolled and assistant nurses. Most (36 out of the 51) had more than five years of experience working with children and adolescents with PHIV. Concerning disclosing HIV to children, 36 out of the 51 reported that they had disclosed to children and adolescents.

### 3.2. Themes

Analysis of the FGDs revealed three main themes and several sub-themes as key findings of this study (Table 2).

#### 3.2.1. Informing Adolescents of their HIV-Positive Status.

##### Deciding the Age to Tell Children

The difficulty of determining the ideal age to initiate disclosure to children emerged as one of the key themes of the focus group discussions. The data revealed that the HCWs perceived age as an important consideration in disclosing to children, even though they had different views about the ideal age of disclosure.


*“I don’t think there is a specific time or age that a child could be ready, but every time the healthcare worker meets the child, they should assess the maturity to determine readiness.” (FDG 4: Nurse).*



*“The important thing is to consider the age of the child…, check if the child is really matured to understand the disclosure.” (FGD 1; Lay counsellor).*



*“Personally, I think at least at the age of 10 years the child has some light and can understand better because this is when puberty starts.” (FGD 1: Social worker).*


##### Deciding Who Should Tell the Child

In the focus groups, the HCWs did not agree on who should tell the child about their HIV-positive status. Some believed that the caregivers should initiate disclosure and provided reasons why they thought so.


*“I say it is the parent..., parents know their children. I think disclosure is the right that should be given to the parent.” (FGD 1: Lay counsellor).*



*“The caregiver should initiate disclosure. The child trust the family more than the nurses at the clinic. Caregiver-initiated disclosure builds trust to the child and helps him/her to believe what the healthcare workers will tell her at the clinic.” (FGD 4: Nurse).*


Others were of the view that the HCWs should disclose and indicated the conditions that would allow the HCWs to initiate disclosure. They referred to children who attend the clinic alone, situations where the caregiver refuses to disclose, or when the caregiver approaches the HCW for assistance with disclosure.


*“The reluctance of parents to disclose to their children leaves us with no option but to disclose to the children ourselves.” (FGD 7: Nurse).*



*“Healthcare workers should disclose because they have more HIV-related information than the caregivers.” (FGD 1: Social worker).*


Others were of the view that the HCWs should disclose because the caregivers provide false information to children during disclosure.


*I can say the healthcare worker because parents sometimes give false information, they don’t explain why or not give all information, they tell children that they will die if they don’t take treatment without the actual information” (FGD 1: Social worker).*



*“I intervened in children who defaulted because they were told false information, they were told that have heart disease. I provide ongoing counselling to support them and make appointments until adherence to treatment” (FGD 1: Dietician).*



*“I was exposed a lot to children who were not disclosed well or lied to by parents. Children were told that they are taking TB treatment and because they start to understand that TB treatment is six months they ask questions why are they still continuing to take such a treatment beyond the six months” (FGD 2: PN).*


At the same time, some argued that disclosure should be a joint responsibility of HCWs and caregivers. They felt that the HCWs would bring skills and expertise to the disclosure process to deal with the emotional reactions of children following disclosure.


*“I think the health worker can initiate disclosure but do it in the presence of the caregiver. During this process, both the caregiver and the health worker should talk, because the role of the health worker is to support the caregiver.” (FGD 7: Nurse).*



*“The caregiver and the health worker should work together during disclosure because it is the health worker who has more information relating to health matters, and is in a better position to explain what will happen if the treatment is not taken as prescribed.” (FGD 8: Nurse).*


#### 3.2.2. The Social Contexts That Influence Disclosure

The HCWs said that the caregivers’ reluctance to disclose the child’s own HIV serostatus was influenced by various factors in the family and community. They pointed out caregiver- and child-related difficulties that influence disclosure to children.

##### Caregivers’ Acceptance of Thier Own HIV Status

The parents’ difficulties in accepting their own HIV serostatus influenced the disclosure of the children’s HIV status. They postpone disclosure to their children because they have not yet dealt with their own HIV status.


*“Sometimes parents do not disclose because they are scared that the child will tell other people about the HIV status, and by so doing reveal his/her status as well as that of the parent; while the parent was not ready for other people to know her HIV status.” (FGD 4; Nurse).*


##### Caregiver’s Knowledge about HIV and Disclosure

The HCWs explained that often caregivers delayed disclosure because they had insufficient HIV-related knowledge and inadequate skills to disclose. They lacked the confidence and emotional preparedness to initiate and handle disclosure and also to answer basic questions about HIV that may arise from the children.


*“Parents are afraid that the child might ask questions about how they contracted HIV. So, parents are scared to be confronted by such questions from children.” (FGD 3: Lay counsellor).*



*“When the mother realises that she will not be able to answer questions about how the child became infected, she then avoids disclosing.” (FGD 8: Nurse).*



*“One parent approached me and told me that she wanted to disclose to her child but did not know how. This child started taking treatment when she was 9 years old and the mother came with her to me when she was 13 years old.” (FGD 1: Social worker).*


##### Child too Young to Understand HIV

The HCWs reported that one of the most common reasons for delaying disclosure to children was the caregiver’s assertion that the child is too young to understand the HIV diagnosis.


*“Parents think that children are too young to understand, generally, they won’t understand hence they don’t disclose.” (FGD 5: Dietician).*



*“Parents do not disclose because they believe that the child is not yet matured enough to understand what is happening regarding HIV (FGD2; lay counsellor).*



*“Parents delay disclosing to their children and say that they are still waiting for child to be matured and ready in order to tell him/her about his/her HIV status (FFGD 6: PN)/*


#### 3.2.3. Fearing the Consequences of Telling

The data revealed that the fear of provoking undesired consequences following disclosure deters caregivers from disclosing to children.

##### Child’s Negative Reaction to Disclosure

The HCWs explained that the fear of possible negative reactions to disclosure was a barrier preventing caregivers from disclosing. They stressed that the caregivers concealed or delayed disclosing HIV status to children to protect them from the negative effects of knowing their HIV status.


*“They think disclosure will lead to children isolating themselves from other people because of the thought that they will die.” (FGD 5: Social worker).*



*“Sometimes the mother is scared that the child may commit suicide.” (FGD 4: Nurse).*



*“Other parents don’t disclose because they fear that they might lose their children, if the child discovers that she was infected through them.” (FGD 7: Nurse).*


##### Concerns That the Child Cannot Keep HIV Secret

The inability of children to keep the HIV diagnosis to themselves was one of the reasons caregivers delayed disclosure. The HCWs stated that caregivers claim that because of the children’s young age, they are incapable of keeping family secrets and would not understand issues of the confidentiality of the HIV diagnosis.


*“The child might go and talk to other children; and the children will be not able to keep that secret because children don’t understand confidentiality.” (FGD 3: Lay counsellor, 34 years).*


##### Subsequent Stigma and Discrimination

The stigma and discrimination against people living with HIV in communities hinder timely disclosure to children. Caregivers delayed disclosing children’s HIV status to them because they fear that their children will be stigmatised, bullied, and isolated at schools and in their communities.


*“The society looks at you with a different eye when you are HIV-positive. The lack of acceptance within homes and society drives those who are HIV-positive to live in constant denial of their condition.” (FGD 7: Nurse).*


##### Being Blamed by the Child

The fear of being blamed by the children for infecting them was also mentioned as a major reason for the delay in disclosing by the caregivers, particularly by the biological mothers. The HCWs believed that mothers live with feelings of guilt as well as fears of being blamed and judged by their children for unknowingly transmitting HIV to them.


*“Parents feel responsible for passing the disease to the children. That’s why they don’t want to disclose.” (FGD 1: Dietitian).*



*“The other thing that causes the parent to fear, is the thinking that the child might hate her for the rest of his/her life, blaming her of infecting him/her.” (FGD 8: Nurse).*



*“I think the parent feels guilty and thinks that the child will blame her for carelessness and for failing to take measures to protect her/him from HIV infection. So, the parent is avoiding to be blamed and accused for the reason that the child is positive.” (FGD 6: Nurse).*


#### 3.2.4. Consequences of Delayed Disclosure

The HCWs referred to the regrettable, negative outcomes of delaying disclosure to older children and adolescents that they had observed in their interaction with children and adolescents with PHIV.

##### Contributes to Treatment Default

The HCWs stated that when children continue to take medication without being informed about their HIV diagnosis they become rebellious or refuse to take treatment or stop attending the clinic completely.


*“Children get tired of taking treatment so they stop taking treatment if they don’t understand the reasons for taking it continuously.” (FGD 2: Nurse).*



*“I have an orphan teenage boy who started taking treatment at birth under the supervision of the grandfather, but when he entered adolescence, he stopped the treatment due to lack of knowledge about his status.” (FGD 4: Lay counsellor).*


##### Poor Clinical Outcomes

Based on the observations and experiences of the HCWs, delaying disclosure contributes to the deterioration of the health status of the children. Defaulting treatment induces treatment failure.


*“Deception about the nature of the disease for children leads to the decline of children’s health status as they become sick due to poor adherence. One child had to be changed to the second line of treatment.” (FGD 1: Dietician).*



*“I realised that unsuppressed viral loads among these children are due to non-adherence of treatment. They sometimes stop taking treatment since they don’t know why they are taking the treatment for.” (FGD 2: Nurse).*


##### Inconsistent and Episodic Adherence to ART


*“I have a case of two orphaned children aged 6 and 13 years old with high viral loads due to non-adherence to treatment. The grandmother explained that they are missing treatment because they come back home very late at night. These children don’t even know why they are taking treatment as no one ever told them about it.” (FGD 7: Nurse).*


##### Children Are Denied a Chance for Self-Care


*“Every time the young one takes her ARVs she also demanded that the older sibling should be given the treatment. She constantly asked why she was the only one taking the treatment. Whenever she fetched her treatment she also distributed to other children in the house and said that they must also drink.” (FGD 8; Nurse).*



*“I discovered during a home visit that he was throwing the tablets under the bed because he was sleeping alone.” (FGD 4: Lay counsellor).*


##### Leads to Accidental Disclosure


*“If the parent does not disclose, the child may discover his/her HIV status accidentally, which may cause denial for a very long time.” (FGD 3: Lay counsellor).*



*“Ooh! Just like what happened to me here. The child has been on ARVs for a very long time. I said to the child, ‘When you are HIV-positive you take this treatment’ and there…, the child started screaming. The child was just crying because he was on ARVs for so long without knowing that he is on HIV treatment. So accidents do happen.” (FGD 1: Nurse).*


##### Risk of Secondary Transmission of HIV


*“The disadvantage of non-disclosure during adolescence is that you may find the child has already started dating and practising unsafe sex.” (FGD 4: Nurse).*



*“Depending on their age, some of these children become sexually active at a very early age, for example at 11 or 12 years, so we do encourage them to use dual protection when engaging in sexual activities.” (FGD 7: Nurse).*


## 4. Discussion

The findings highlight the contextual challenges faced by caregivers in disclosing the child’s own HIV diagnosis and the negative consequences of delaying disclosure on the psychosocial health of the children, as perceived by the HCWs providing HIV treatment and care to them. The study found that although researchers consider caregivers to have the principal role of telling their children about their HIV diagnosis [8,24,40], the HCWs in the current study did not agree on who should tell the child about their own HIV-positive status. They indicated that disclosure to children with PHIV should be a joint responsibility of the HCWs and caregivers. The findings are consistent with those of previous studies [5,6,20,30,41]. Studies done among caregivers of children with PHIV revealed that they were of the same view—that disclosure should be a joint responsibility [27,42].

Research has reported different results regarding the ideal time at which HIV disclosure to children should begin [8,20,29,40]. That observation is in line with the finding of the current study, which is that the ideal age to disclose to children is debatable. The HCWs did not reach a consensus. Kalembo et al. [20] attribute the variation in the opinions about the ideal age of disclosure to children to a lack of standardised disclosure protocols in SSA. The HCWs in this study, as in others, believed that disclosure should be done when the child is mature enough to understand the implication of an HIV diagnosis. In the current study, 40 of the 51 participants felt that disclosure should happen when the child is between 10 and 12 years old because early and timely disclosure is beneficial to children. Similarly, caregivers in prior studies disclosed when they felt that the child was old or matured enough [1,24,27]. Madiba and Mokgatle [8] suggest that the recommended ideal age of disclosure is likely to be influenced by the community and social contexts of disclosure. Nonetheless, the WHO’s recommended age to complete the disclosure process to children with PHIV is before the age of 12 years.

The HCWs reported that the caregivers’ decision to withhold or delay disclosure to the child or adolescent was influenced by caregiver- and child-related contextual and social factors. Other researchers have documented complex and difficult social environments for disclosure at the caregiver-child dyad, and at the community level (such as neighbours, friends, schools, churches and the media) [4,7,8,20,22,28,29]. The current study has found that the caregivers’ social contexts were the main barrier against informing the children timeously about their HIV diagnosis. The fear of being blamed, the fear that the child will not maintain the secrecy of the HIV status, and the fear of subsequent stigma and negative discrimination against the entire family largely influenced the timing of the disclosure. Other studies conducted with caregivers have reported that the high levels of internalised HIV stigma and shame on the part of biological mothers living with HIV induced them to delay disclosure to their children [21,42,43,44,45,46]. Reducing the stigma associated with HIV infection is critical in this setting to create a safer and more supportive environment for the timely and safe disclosure to children [11,45].

The HCWs observed that the caregivers’ limited disclosure skills and inability to answer children’s questions about HIV in an accurate, age-appropriate manner hindered the early and timely initiation of disclosure to children. They believed that caregivers, particularly biological mothers, are reluctant to disclose because their children might ask questions pertaining to how they became infected. As such, they harbour deep-seated feelings of guilt about transmitting HIV to their children and are afraid of experiencing subsequent blame and rejection by their children. The HCWs believed that mothers fear being considered by their children to have been promiscuous, following the disclosure of their children’s HIV serostatus. Research suggests that caregivers who lack the skill to handle the disclosure process positively opt for silence, in the process delaying disclosure to later in their children’s adolescence [7,11,12,22,24,27,29,43,46]. HIV-positive mothers who have not accepted their own status and struggle with coping find it more difficult to disclose the child’s status because of discomfort with the issue of their own HIV status. Similar findings have been reported elsewhere [21,22,24,40].

Other barriers to timely disclosure included child-related factors such as caregivers’ fear of the potential negative reactions of the children upon learning their HIV status, feeling that the child is not ready to learn his/her HIV status, assertions that the child is too young to comprehend the meaning and consequences of an HIV diagnosis, and fear that knowledge of his/her HIV status would traumatise the child [21,22,40,42,43,45]. According to the HCWs in this study, disclosure is more complex when the caregivers have multiple children in a household, some of whom are HIV-infected and others who are not. This phenomenon has been observed in previous studies [45]. Mweemba et al. [22] suggest that barriers against disclosure to adolescents are not mutually exclusive but are interrelated, and that caregivers, therefore, need support or capacity-building in respect of the process of HIV disclosure. This underscores the importance of the involvement of HCWs in the disclosure process to children and adolescents with PHIV.

The HCWs identified key negative consequences of delaying disclosure to children and adolescents with PHIV. Delayed disclosure denies children the opportunity to make correct and informed decisions about their health and contributes to children defaulting or refusing to take medication, which can lead to the development of a drug-resistant strain. Delayed disclosure might result in transmitting HIV to potential sexual partners and others, and results in unsafe sex practices among older adolescents with PHIV. These findings are compatible with those of other studies that documented the serious consequences of delaying informing the child about his/her HIV status [7,32,42,47,48].

The study revealed that the most important consequence of delayed disclosure is poor adherence to treatment, with a negative impact on the child’s health outcomes, including treatment failure as a result of treatment default. This finding is similar to those of other studies that linked non-adherence to ART among children due to the children’s lack of knowledge of their HIV-positive status, and lack of understanding of the consequences of defaulting [7,8,27,49]. The HCWs stressed that adolescents with PHIV become tired of taking treatment and refuse to take it or stop taking it when information about their disease is not provided. During disclosure, the HCWs would not only tell the child about the diagnosis but also explain the disease and its outcome entirely.

Other studies associate delayed disclosure with feelings of anger and resentment, particularly when disclosure occurs very late into adolescence [3]. Mburu et al. [25] found that delayed disclosure often creates tensions and emotional difficulties for adolescents. In a disclosure intervention in Botswana, the researchers reported negative impacts of prolonged non-disclosure, such as rebelliousness and angry confrontations with parents [7]. Anger can be directed to the caregiver. This is typical when there is mistrust of caregivers, because of the lies that were told before disclosure [34,43,45]. Research shows that anger is intensified when children were given untrue information about why they had to take their medicines before being told about their HIV diagnosis [7,12]. Prior to their HIV disclosure, children are exposed to numerous often negative messages about HIV, which is presented as a fatal sexually transmitted disease [7,30,45]. Adolescents’ resentment of the HIV diagnosis leads to their missing doses or stopping ART altogether [7,9,33,34,50].

Consistent with the current study, research suggests that delayed disclosure can lead to accidental disclosure. The HCWs narrated incidents of children discovering their HIV diagnosis accidentally in health facilities by reading clinical notes, overhearing conversations among HCWs and caregivers, and accessing information about HIV and treatment from the media. The HCWs were of the opinion that children become prone to accidental disclosure due to a lack of adequate measures and practices to prevent such events in facilities. In addition, children and adolescents may discover their HIV status through their own gradual realisation related to their need to take drugs [41]. Some studies indicate that a child might be informed about his/her HIV diagnosis by a family member before the official disclosure [1,2]. Accidental disclosure is undesirable because children who experience it feel betrayed, become violent and rebellious, are prone to commit suicide, and have irreversibly spoiled relationships with their caregivers [30].

As with other studies, this study has found that delaying informing children of their HIV diagnosis compromises their independence and self-responsibility, and leads to poor self-care behaviours such as poor adherence to medication and risky sexual behaviours. When children are unaware of their HIV diagnosis, they are deprived of the opportunity to participate in their own health care [6,29,30]. Our findings and those of other studies suggest that the timely and safe disclosure of HIV status is an essential component in allowing the involvement of the adolescent in his/her self-care and a smooth transition to adulthood [11,51]. The interaction between HCWs and children improves when children are aware of their HIV diagnosis. Continuous interaction between HCWs and children with PHIV has been shown to facilitate the children’s acceptance of their condition [29] and is crucial in helping them to understand and deal with the potential consequences of their HIV status [12].

The HCWs reported that a delay in disclosure might result in children unknowingly transmitting HIV to their sexual partners once they start indulging in sexual activities. As undisclosed adolescents are not particularly motivated to prevent transmission through safe sexual behaviours, there is a persistent risk of transmitting a resistant strain of the virus. This has been reported in several studies [8,9,20,30,52]. Earlier and safe HIV disclosure reduces the risk of the secondary transmission of HIV by equipping children and adolescents with information on how HIV is transmitted [8,11,30]. Awareness of their own HIV status allows adolescents to make appropriate choices regarding their sexuality in relation to their status.

The results of this study are based on FGDs with a small number of HCWs. Their opinions are not necessarily representative of the opinions of all the healthcare service providers of this region. Therefore, the findings cannot be generalised. One other limitation is that the focus group comprised different professional categories and types of HCWs, this might have influenced the participants’ responses during the discussions. To mitigate, the researcher probed and made follow-ups to get more information and clarity particularly from participants who were quiet and brief in their responses. The gap that the current study did not address was to examine the influence of the professional background, years of experience working with children with PHIV, and training on the practice of disclosure. Large studies employing mixed methods studies will fill this gap.

## 5. Conclusions

The study findings suggest that delaying disclosure to older children and adolescents contributes to children defaulting or refusing to take their medication and poor self-care behaviours. Additionally, delayed disclosure gives rise to anger and resentment and could lead to accidental disclosure. Disclosing early and in a timely manner helps children and adolescents to understand their health status, enables their active participation in self-care and treatment, and enables them to accept their HIV status. Knowing their own HIV serostatus is important for children and adolescents, making it possible for them to attain independence in caring for themselves and motivating them to take their medications consistently. It is imperative to train HCWs in the skill of supporting caregivers and children through the disclosure process, to ensure that the benefits of disclosure are realised by the children and their caregivers

This study revealed that the barriers preventing the disclosure of HIV status to children are not mutually exclusive, and that caregivers’ social context is the main barrier to inform children about their HIV diagnosis. Their high levels of internalised HIV stigma are the chief reason for their delaying disclosure to their children. Reducing the stigma associated with HIV infection is critical in this setting, to create a safer and more supportive environment for timely and safe serostatus disclosure to children. While early, safe, and timely disclosure is beneficial to children, any strategies to encourage caregivers to disclose early should take into consideration the concerns caregivers have about the negative impacts of disclosure.

## Figures and Tables

**Table 1 children-07-00289-t001:** Demographic information of study participants.

Variables	Characteristics	Frequency (%)
Sex	Female	46 (90.2)
	Male	5 (9.8)
Designations of HCWs	Lay counsellor	16 (31.3)
	Enrolled nursing assistant	1 (1)
	Enrolled nurse	2 (3.9)
	Professional nurse	24 (47)
	Social worker	5 (9.8)
	Dietician	3 (5.9)
Age group	20–29	5 (9.8)
	30–39	15 (29.4)
	40–49	16 (31.4)
	50–59	14 (27.4)
	60–69	1 (2)
Paediatric HIV experience	<1 year	4 (7.8)
	1–4 years	11 (21.6)
	5–10 years	19 (37.3)
	>10 years	17 (33.3)
Ever disclosed HIV to a child?	Yes	36 (70.6)
	No	15 (29.4)
Ideal age of disclosure	8–12 years	40 (80)
	<12 years	2 (4)
	>8 years	8 (16)
HCW who have disclosed	Lay counsellor	13 out of 16
	Enrolled nursing assistant	0 out of 1
	Enrolled nurse	0 out of 2
	Professional nurse	19 out of 24
	Social worker	3 out of 5
	Dietician	1 out of 2

**Table 2 children-07-00289-t002:** Themes and subthemes identified from the focus group discussions.

Informing Children of Their HIV-Positive Status	Deciding the Age to Tell the Child
	Deciding Who Should Tell the Child
The social contexts that influence disclosure to children	Caregiver and child characteristic The caregiver has not accepted own HIV status Caregiver knowledge about HIV and disclosure Child too young to understand HIV
	Fearing the consequences of telling Child’s negative reaction to disclosure Concerns that the child cannot keep HIV secret Subsequent stigma and discrimination Being blamed by the child
Consequences of delayed disclosure	Contributes to treatment default Inconsistent and episodic adherence to ART Poor clinical outcomes Children are denied a chance for self-care Leads to accidental disclosure Risk of secondary transmission of HIV

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
