# Peer review of "The Consequences of Delaying Telling Children with Perinatal HIV About Their Diagnosis as Perceived by Healthcare Workers in the Eastern Cape; A Qualitative Study"

_children, 2020, doi:10.3390/children7120289_

Round 1

Reviewer 1 Report

Overall comments

The present study investigates the social and contextual barriers to disclosing perinatal HIV infection to children in areas of the Alfred Nzo district of Eastern Cape province in South Africa. Previous studies in various communities have shown that delaying HIV disclosure into adolescence is associated with increased community transmission and non-adherence/failure of antiretroviral therapy, as well as psychological, behavioural, and other issues in young carriers. To identify factors influencing delayed disclosure by healthcare workers (HCWs) and caregivers, the authors conducted focus group discussions with 51 HCWs and performed thematic analysis.

While the new ground broken by the present study is limited, it does capture an important snapshot of community attitudes towards HIV disclosure to children/adolescents in one region of Sub-Saharan Africa. To this end, the title accurately reflects the core elements of the study. The study appears to have been conducted ethically and with the informed consent of all participants. Related studies are drawn upon in the introduction and discussion to contextualize the present study, which is of comparable size and scope to similar articles published in this journal. The manuscript is generally clear, although I have a number of suggestions in this regard which I hope may be of value. Thank you for the opportunity to read and review the present work.

Major improvements

  1. Figure 1. In the discussion, various factors are identified as major barriers to HIV disclosure, negative consequences of non-disclosure, and so on. Instead of merely listing these factors in Figure 1, I feel that ranking each of these factors, from most important at the top of each box to least important, could maximise the usefulness of this figure for readers.
  2. Methods, lines 93-107. Was there any particular reason for selecting the sampled areas in the Alfred Nzo district? For example, is the incidence of HIV infection in this region similar/dissimilar to SSA in general? Are there other important demographics that were taken into account in the study design or should be considered by the reader?
  3. Discussion, final paragraph. The inability to generalize the findings of the present study is listed as its singular limitation. While this is a key limitation, are there no other limitations, for example, to thematic analysis itself? At least some focus group participants appear to have known other participants in the study; could personal, professional or power relationships be a confounding factor in participants’ answers during group sessions? Do the results in the present study indicate a need for further research or prompt additional research questions? Please add more detail to this paragraph.

Minor improvements

  1. Various places. Please define all abbreviations used in the first instance. E.g., line 2, ‘HIV’ as ‘human immunodeficiency virus’; line 42, ‘SSA’ as ‘Sub-Saharan Africa’, etc.
  2. Line 44. Please specify who recommends disclosing HIV status to children before the age of 12 (e.g., “While the World Health Organization recommends disclosing…”).
  3. Methods, line 131. Please state the version of NVivo software used.
  4. Lines 142-150. I question whether most of this paragraph is strictly necessary. Key elements of the methodology (e.g., verbatim transcriptions, use of NVivo software, etc) could be extracted and merged with the previous paragraph.
  5. Figure 1. Please ensure consistent formatting between each box (i.e., fonts, bullet points and box colours). Please also ensure consistent grammar is used in each point as well as the titles (e.g., passive / active voice).
  6. Figure 1. The different colours do not seem to aid the clarity of the figure or denote crucial information. I suggest using three different shades of the same colour or grayscale.
  7. Figure 1, final box. “Increased risk for of” should read “Increased risk of”.
  8. Figure 1, caption. “Themes and subthemes” is not specific enough. If these were the themes identified from the focus group discussions (as suggested in the main text), this should be more clearly specified.
  9. Reference 11. The number 11 is listed twice. Please revise.
  10. Reference 14. Reference is not a journal article, but the current formatting is for a journal article. Please revise.

Author Response

We thank the reviewer for the valuable comments to improve the clarity of our manuscript.

We thank the reviewer for the valuable comments to improve the clarity of our manuscript. We have outlined the responses to the comments and suggestions in the attached document.

Reviewer 2 Report

The manuscript of Madiba et al. reported a qualitative research to investigate the consequence of delaying disclosure of HIV status to the children or adolescent, and identify the main barrier. They have revealed three main themes and some sub-themes as the factors that will delay the disclosure, and described the potential increased risk for those who did not received timely disclosure of HIV status. Generally, the manuscript is clear, and well written; however, several concerns should be addressed before this can be considered further publication.

  1. The major concern is whether the results from the research can support the conclusion. Since this is a qualitative study, it can be used to identify those factors related to the barriers or find risky consequences reported by the HCWs; however, in the manuscript, it is concluded the delaying disclosure will increase the risk, which is not well founded by the data. When talking about increased risk, there should be detailed numbers and the following statistical analysis to determine whether the risk is increased. The author may consider changing the conclusion or adding more results, such as an objective scoring test.
  2. The demographic background of HCW are diverse based on table 1. It will be interesting to see whether their background will affect their report or input for the different themes.
  3. In the section of results, for each theme that was identified, did a consensus reached among the HCWs or just a portion of them? It is not clear the threshold to propose a theme from the FGD or if these themes were included in the guide for discussion.
  4. Another limitation of the study is that the conclusions were only based on reports from HCWs, but lacking the caregiver’s perception. Therefore, the bias may exist.
  5. In the discussion, the best disclosure timing of disclosure agreed by the HCWs is between 10 and 12 years old. But the concern is whether this is an ideal age should not be only determined by the observations from HCWs, but also need more comparisons among different age groups to reach, which should be discussed. This is an example to show that although inputs from HCWs are important, it does not mean what they report must be correct. It should be cautious to make conclusions from the results. 

Author Response

We thank the reviewer for the valuable comments to improve the clarity of our manuscript. We have addressed the comments and suggestions in the relevant sections of the manuscript as outlined in the attached document.

Reviewer 3 Report

This manuscript is based on an important topic and the authors have done a great job in conducting such kinds of studies. However, the following description of the major issues identified in this manuscript:

  1. Please provide a concise and attractive title. I strongly feel that title is too lengthy.
  2. The introduction parts need to be shortened. Please consider shortening. For example line, no 41-82 can be shortened and can be improved by removing the repetition.
  3. In the introduction section, Please provide the estimated number of children living in the study site. Also if possible please include estimate no. of children who are actively undergoing ART program.
  4. Please move the line no 84-89 in the conclusion section. These sentences do not fit in the introduction section.
  5. Please improved the figure as the current figure doesn’t clearly show the main theme and sub-theme identified from this study. I feel there is an overlapping under the main theme “The social contexts that influence disclosure” if I am not wrong. There is a repetition and could you please explain these? Please reconstruct a new figure.
  6. The legend of the figure should be able to explain the figure. Please redraft the figure legends in a better way.
  7. In the result section, the subtheme “child is too young to understand” should go under the theme  “Informing adolescents of their HIV-positive status” and should also be merged with the subtheme Deciding the age to tell”
  8. There should be a comparison in the experience and opinion of the HCWs who disclose HIV status to child and HCWs who never disclose the status to a child. Please consider including these data in the manuscript.
  9. I feel that data could be beneficial to the readers if this study could also provide who is the right person (HCWs/caregivers/parents or all) to disclose the seropositive status to the child based on the previous experience from the HCWs who disclose the HIV status to a child earlier.
  10. The subtheme “the child is too young to disclose” should go under the theme “ deciding age to tell”
  11. In the discussion section, Line no. 337-338, “The participants in the current study felt strongly that disclosure should happen when the child is between 10 and 12 years old”. Please mention exactly how many participants strongly felt that. Otherwise, it’s a vague or assumption.
  12. Line no 338-339, similarly, caregivers in prior studies based on the timing of disclosure on the child’s age and maturity”. This particular statement is unclear and doesn’t make any sense. Please redraft and correct the sentence.

Author Response

We appreciate the time the reviewer took to review our manuscript and we have addressed the comments in the relevant sections of the manuscript as highlighted in the attached document.

Reviewer 4 Report

Overall, I think that this is a well-written manuscript in that the ideas flow easily from point to point.  The manuscript addresses an important issue, when and how to tell a child about his/her HIV status by soliciting focus group data from healthcare workers who work with seropositive children and their families.

There are, however, a number of issues that should be addressed.  

1.  The easiest issue to fix is the occasional typo and word usage issues.  For example, on Line 76, it should be "adolescents" with an "s" on the end. On Line 194, the "s" in HCWs should not be capitalized.  Line 359, it should be "...feelings of guilt ABOUT transmitting...".  On Line 160 it should be "...sample WAS comprised of..."

And while I learned that "timeously" is in fact a word, I suggest you rephrase the two times (Line 347 and 451) you use it.

On Line 111 and elsewhere, phrasing such as "their HIV status to children..." is a bit awkward as it suggests that the parent is disclosing their own HIV status (the parents) to their child.  

2.  On p. 3 you note that there were 8 focus groups (Line 109), but say that there were only 7 on the following page (Line 159).

3.  In the Methods, briefly describe how many people were approached to participate in the study and how many declined.

4.  Figure 1 has "adolescents" in the title, but then the content refers to child and children.  Be consistent.

5.  All of the above are fairly straightforward fixes.  Where the manuscript runs into issues for me is in the Discussion.  Specifically, you make a number of statements that are not supported by the study design.  For example, on Line 387 you note that a consequence of delayed exposure is poor adherence to treatment.  Problems with this statement include:  this is based on HCWs report, not the children themselves.  And while non-disclosure is not ideal, there could be any number of factors that lead to poor adherence, such as lack of funds for medication and general adolescent rebellion about being told what to do. 

Basically, to know how non-disclosure specifically impacts the child and adolescent, you would need to ask them directly.  Similarly with parents' internalized HIV stigma.  It's probably an issue, but the HCWs are not in the best position to attest to that.

Somewhat minor -- Depending on the nature of the parent-child relationship pre-disclosure, my guess is that "irreversibly spoiled relationships with their caregivers" is the exception, not the rule.

6.  One question I had while reading the manuscript, that you can't know, is "What are the children told about their condition?"  They take medication on a regular basis, see medical professionals regularly, so they must be told something.  My point being that depending on what they are told in the beginning seems particularly relevant to how they respond once they are told the complete truth.  If it was always framed as a serious illness, they may be less likely to be upset than if they were told something to minimize the situation initially.

7.  Somewhat related to that, the study seems centered on when is the best time to tell a child the truth and who should do it.  But it seems that it really should be about how much to tell a child at each stage of their development in order for them to best process the diagnosis.  A young child probably need only know that s/he is sick and needs medicine regularly.  As a child gets older, then it seems appropriate to have additional conversations.  I see it as an ongoing process until the child is emotionally and cognitively able to fully process their diagnosis.

8.  The Limitations section is very under-developed.  One of the things that should be mentioned is that you only have HCWs side of "the story."  You do not have input from parents of HIV+ children nor of children who have since found out their status, which would help flesh out consequences of non-disclosure, how parent-child relationships were affected, etc.

Author Response

(The authors gave the same response as above.)

Round 2

Reviewer 2 Report

The manuscript has been improved comparing to its original version. The language is clear, and the findings are clearly stated.Therefore it is suitable for acceptance. 

Author Response

We thank the reviewer for their comments

Reviewer 3 Report

The authors were responsive to reviewers' comments and have improved the manuscript. Most of the concerns have been addressed in a satisfactory manner. Congrats to the authors.

Author Response

We thank the reviewer for their comments. 

Reviewer 4 Report

This is a revision of a manuscript on healthcare workers' (HCWs) perceptions of consequences related to delaying disclosure to children with HIV.  This is an interesting topic that has the potential to influence the ways in which HCWs are trained to help parents/caregivers navigate disclosure.  There are, however, a number of issues  that need to be addressed.

The main issue is that the authors frequently present data based on HCWs' perceptions as fact. Based on focus groups with HCWs (rather than the parents or children in question), it cannot be stated, for example, that "parents' difficulties in accepting their own serostatus influenced the disclosure of the children's HIV status" (Lines 218-219).  This is a causal statement that would need to be based on (a) interviews with the parents themselves or (b) questionnaires administered to the parents. That sentence represents just one of many such examples. The data are HCWs beliefs not universal truths.

Qualitative research can be very informative, but it also limits the ways in which results are presented.  For example, it is difficult to make the case that "one of the most common reasons for delaying disclosure" was child age (Line 237), when there is only one quote to support that belief.  How many HCWs noted that as an issue?  If more than 50% did, then you could probably make such a statement.

Mothers' guilt about transmitting the disease to their child/children is mentioned a few times.  Without direct inquiry of the mothers, at best this is just the HCWs' opinion.  At worst, it is the HCWs' belief that the mothers should feel guilty.

The quotes under the heading "Children are denied a chance of self-care" don't fit that heading.  In the first example, the child is taking the medicine and wants to share it with siblings. It's potentially harmful to the siblings, but how is that denying self-care?  The second quote is about a child discarding his meds.  It's difficult to know why that was done, but it does not necessarily mean a denial of an opportunity for self-care.

One of the quotes under the "accidental disclosure" sections seems a bit problematic because it was the HCW who accidentally disclosed.  That shows lack of professionalism on their part, not shortcomings from the parent that might have been avoided had the parent disclosed.

You state in the Discussion that HCWs stated that it was the joint responsibility of HCWs and parents to disclose to the child.  According to the Results, some HCWs felt this way, others did not.  Therefore, you can't present this statement in a way to suggest that there was 100% agreement about joint involvement in disclosure.

While non-disclosure is not ideal, you err on the other extreme by making statements that disclosure will lead to a range of positive outcomes.  It does not follow that once a child is told about his/her status that medication adherence will improve, be consistent.  In addition, there are multiple barriers to safe-sex practices.  Knowing one's status is not a guarantee that precautions will be taken.  Basically, a more nuanced discussion of the benefits of disclosure is needed. 

A couple of places in the Discussion you present new information that should have been included in the Results.  For example, on Lines 412-413 you name other ways that children learned about their diagnosis.  These examples should be first described in the Results.

Minor Issues:

1.  You switch between "children," "children and adolescents," and "adolescents" throughout the paper.  And it is unclear if when you do so, you are making statement that are specific to only the group named or are just being inconsistent with terminology.

2.  The acronym PHIV is not needed in the Abstract as it is not used subsequently in that section of the paper.

3.  Reference(s) needed for the sentence that ends on Line 35.

4.  Line 23:  Does "they" refer to the caregivers, children, or both?

5.  Line 31:  HIV before (PHIV) is redundant.

6.  Line 84:  There is a word missing between "rural" and "with."

7.  Line 120:  Which language is CD fluent in?  The focus groups were conducted in two, correct?  But you only note that CD is fluent in the language, singular.

8.  Be consistent -- are they focus group discussions or interviews?  The latter term, which is used frequently, suggests one-on-one sessions.

9.  Line 148:  You note that some demographic features are listed in the Table.  What was excluded?

10.  Lines 160 and 162:  Is it a figure or a table?

11.  Line 173:  Is the word "light" in the quote correct or a typo?  

12.  If age is in fact that most common reason why caregivers don't disclose, why is there only one quote to illustrate this point?  It could be that the quote selected did the best job of illustrating the point.  If so, make that clear.

13.  Line 282:  And elsewhere, the use of the word "default" is a bit awkward.

14.  Lines 283-285 would benefit from rephrasing.  There could be reasons other than non-disclosure for children becoming rebellious and refusing meds.

15.  Line 298:  Should the word in the quote be "why" or "what"?  It could be that the interviewee misspoke or it could be a typo.

Author Response

We made corrections to all the minor comments raised as outlined in the document and the relevant sections in the document. In the response to the comments raised in the first review, we feel that we responded line by line to the comments raised and explained the role of the HCWs disclosure to children with PHIV in an extensive and elaborate statement. The comments raised in this second review are based on the same principles about HCWs and disclosure.

We have added references to indicate studies conducted with HCWs on disclosure to children with PHIV in many settings, we trust that this will assist us in explaining the position and the premise of the manuscript. 
